# Sympathetic motor neuron dysfunction is a missing link in age-associated sympathetic overactivity

**Lizbeth de La Cruz[1], Derek Bui[1], Claudia M Moreno[1,2], Oscar Vivas[1,3]***

[1]Department of Physiology and Biophysics, University of Washington, Seattle, United States; [2]Howard Hughes Medical Institute, Chevy Chase, United States; [3]Department of Pharmacology, University of Washington, Seattle, United States

## eLife assessment

This **important** study describes changes in excitability in motor neurons of the peripheral autonomous nervous system during aging. The article provides **convincing** evidence indicating that sympathetic neurons from aged mice show higher excitability compared to neurons from young mice which was linked to decreased activity of KCNQ2/3 potassium channels. This research has implications for understanding the age-related changes that occur in the peripheral nervous system.

**\*For correspondence:**
vivas@uw.edu

**Competing interest:** The authors declare that no competing interests exist.

**Abstract** Overactivity of the sympathetic nervous system is a hallmark of aging. The cellular mechanisms behind this overactivity remain poorly understood, with most attention paid to likely central nervous system components. In this work, we hypothesized that aging also affects the function of motor neurons in the peripheral sympathetic ganglia. To test this hypothesis, we compared the electrophysiological responses and ion-channel activity of neurons isolated from the superior cervical ganglia of young (12 weeks), middle-aged (64 weeks), and old (115 weeks) mice. These approaches showed that aging does impact the intrinsic properties of sympathetic motor neurons, increasing spontaneous and evoked firing responses. A reduction of M current emerged as a major contributor to age-related hyperexcitability. Thus, it is essential to consider the effect of aging on motor components of the sympathetic reflex as a crucial part of the mechanism involved in sympathetic overactivity.

## Introduction

This study focuses on the mechanisms underlying the deterioration of the sympathetic nervous system as we age. With advancing age, the sympathetic nervous system tends to become overactive, evidenced by increased electrical activity in sympathetic nerves (*Wallin et al., 1974*; *Ito et al., 1986*; *Narkiewicz et al., 2005*; *Hart et al., 2009*). This overactivity, observed in humans and animal models, leads to heightened release of norepinephrine over organs (*Ziegler et al., 1976*; *Goldstein et al., 1983a*; *Goldstein et al., 1983b*; *Veith et al., 1986*; *Esler et al., 1995*), triggering compensatory processes and cellular deterioration (*Hogikyan and Supiano, 1994*). Understanding the physiology and pathophysiology of this age-associated overactivity is crucial as many common age-driven diseases are linked to sympathetic overactivity. For instance, dysregulation of norepinephrine release is associated with age-related hypertension and arrhythmias. However, the neuronal mechanisms underlying the overactivity of the sympathetic nervous system remain to be elucidated.

During execution of sympathetic reflexes, three essential cellular components of the nervous system come into play: sensory neurons for detecting and transmitting signals of internal conditions,

**Figure 1.** Components of the sympathetic reflex. Schematic of the components of the sympathetic autonomic reflex: Sensory neurons send information to the central component (pre-ganglionic neurons), where information is processed. Then, sympathetic motor neurons receive information from pre-ganglionic neurons to transmit it to the target organ. This research was focused on evaluating the age-related changes in the function of sympathetic motor neurons.

neurons of the sympathetic nuclei in the brain for integrating information and responding to internal changes, and the sympathetic motor neurons in the ganglia for receiving and delivering information to target organs (*Figure 1*). Age-associated overactivity of the sympathetic nervous system has been predominantly ascribed to changes in the central system, specifically focusing on the hypothalamic and brainstem nuclei. In the paraventricular nucleus of the hypothalamus, the activation of glutamatergic neurons through leptin pathways has emerged as a significant contributor to sympathetic overactivity (*Shi et al., 2015*; *Shi et al., 2020*). At the same time, glial senescence and subsequent inflammatory mechanisms in brainstem nuclei also have been linked to sympathetic nervous system overactivity during aging (*Balasubramanian et al., 2021*). However, one key question remains: Is age-related sympathetic overactivity limited solely to changes in the central system?

Aging impacts the morphology and calcium responses of sympathetic motor neurons, the peripheral component of the sympathetic reflex. Notably, sympathetic innervation to the spleen and lymph nodes reduces, while thymus innervation increases with age (*Madden et al., 1997*). Similarly, innervation to vascular smooth muscle changes with age, and the extent and implications of these changes depend on the specific target and neuropeptide content (*Burnstock, 1990*; *Andrews and Cowen, 1994*; *Andrews et al., 1996*). Additionally, aging affects the function of sympathetic motor neurons, as evidenced by altered calcium responses. Old sympathetic motor neurons display reduced expression of SERCA pumps, an essential component for calcium transport into the endoplasmic reticulum (*Pottorf et al., 2000*; *Buchholz et al., 2007*). The results of these experiments suggest that aging has a distinct influence on the intrinsic function of sympathetic motor neurons, irrespective of any changes in the central system. As a result, the main objective of our study was to examine directly the impact of aging on the intrinsic membrane electrical properties of sympathetic motor neurons (*Figure 1*).

Our work was structured around three main objectives: (1) to standardize the isolation of sympathetic motor neurons from adult to old age in mice, (2) to compare the spontaneous and evoked electrical activity of sympathetic motor neurons at three life stages, and (3) to identify potential molecular candidates underlying the observed altered electric properties and neuronal activity.

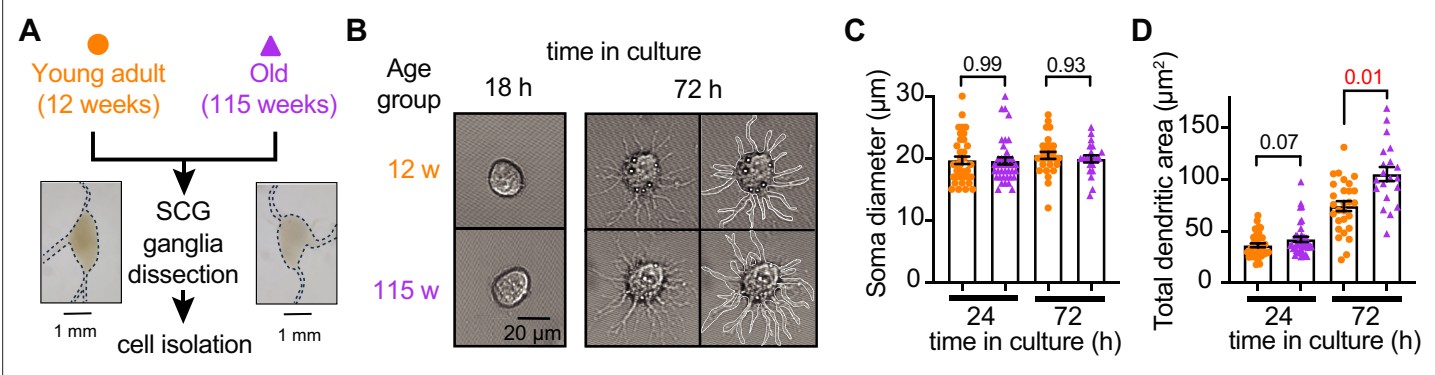

**Figure 2.** Sympathetic motor neurons from old mice are healthy in culture. (**A**) Diagram of the experimental approach: Sympathetic motor neurons were isolated from the superior cervical ganglia (SCG) of 12- and 115-week-old mice. The ganglia did not show morphological differences between ages. (**B**) Differential interference contrast images of sympathetic motor neurons in culture for 18 hr and 72 hr. Right images at 72 hr outline the dendritic area measured. w, weeks. (**C**) Comparison of soma diameter between neurons isolated from 12 or 115 weeks of age at two time points in culture. Orange circles showed single cells from 12-week-old mice while purple triangles show single cells from 115-week-old mice. (**D**) Comparison of the total dendritic area as a proxy for neurite regeneration in culture. Data points are from N = 3 animals, n = 38 cells, from 12 weeks old, and N = 3 animals, n = 39 cells, from 115 weeks old. Error bars represent SEM. p-Values are shown at the top of the graphs. Red values indicate p-values<0.05 while black values indicate p-values>0.05.

## Results

### Sympathetic motor neurons from old mice are healthy in culture

Our first goal was to assess the viability of neurons isolated from young adult and old mice. We isolated sympathetic motor neurons enzymatically from the left and right superior cervical ganglia (SCG) of young adults (12 weeks old) and old animals (115 weeks old, *Figure 2A*). The same enzymatic digestion was used for both ages. Single neurons exhibited no visible neurite growth before 18 hr (*Figure 2B*, left); however, after 72 hr in culture, both young and old sympathetic motor neurons showed evident neurite growth (*Figure 2B*, right). The mean diameter of the cell soma was found to be similar in young ($19.7 \pm 0.6$ µm) and old ($19.6 \pm 0.6$ µm) neurons after 24 and 72 hr in culture (*Figure 2C*). The dendritic arborization after 24 hr in culture was comparable between young ($37 \pm 2$ µm²) and old ($42 \pm 3$ µm²) neurons. However, after 72 hr in culture, the dendritic arborization in old neurons ($105 \pm 7$ µm²) was more extensive compared to that of young neurons ($74 \pm 5$ µm²) (*Figure 2D*). We continued the culture for 7 days and noted that the neurites grew until they contacted other neurons in the same dish. Furthermore, glial cells became apparent at this stage in cultures of young and old cells. The ability of neurons to regenerate neurites and contact each other was considered an indicator of viability and health. We ruled out the possibility that any functional differences between neurons from animals of different ages were due to potential damage to old neurons during the isolation and culture process. From now on, we report electrical measurements on neurons that had been incubated for approximately just 12–18 hr after isolation.

### Old sympathetic motor neurons fire action potentials spontaneously

To investigate whether aging alters the function of sympathetic motor neurons, we first compared the spontaneous activity and passive electrical properties of neurons isolated from mice at different ages: 12 weeks (young), 64 weeks (middle age), and 115 weeks (old). For reference, we also provide comparable human ages (*Figure 3A*). Spontaneous activity was recorded using the current clamp modality without applying any holding or current stimulus. Electrical access to the cell was obtained using the perforated patch technique (see 'Materials and methods' for details). Young neurons had a resting membrane potential (RMP) of $-64 \pm 1$ mV and very little spontaneous activity, which aligns with the expected behavior of motor neurons that respond to presynaptic commands (*Figure 3B and C*). Only 3% exhibited spontaneous firing (*Figure 3D*). In contrast, middle-aged neurons showed an RMP of $-58 \pm 1$ mV, and 37% of these neurons displayed spontaneous firing (*Figure 3B–D*). The depolarized RMP and the increased number of neurons spontaneously firing persisted in old neurons, where the RMP was $-54 \pm 1$ mV and 58% of neurons exhibited spontaneous firing (*Figure 3B–D*). Despite the

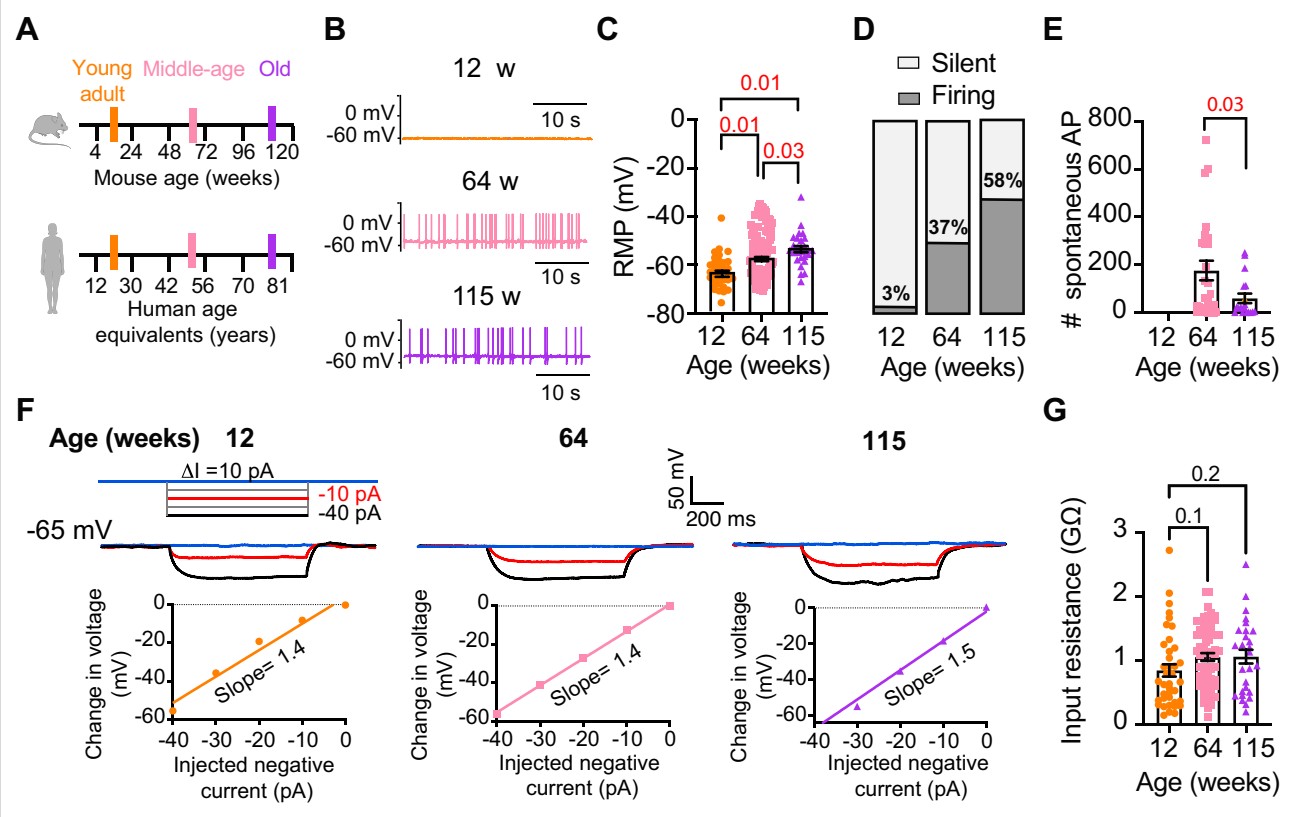

**Figure 3.** Sympathetic motor neurons from old mice fire action potentials spontaneously. (**A**) Schematic of ages in mice, and equivalent in humans, that were used to compare the functional responses of sympathetic motor neurons. (**B**) Representative membrane potential recordings of spontaneous activity from neurons isolated from 12-, 64-, and 115-week-old mice. (**C**) Comparison of the resting membrane potential (RMP) between different ages. (**D**) Comparison of the percentage of silent and firing neurons between different ages. (**E**) Comparison of the number of action potentials (APs) fired spontaneously in 1 min between different ages. (**F**) Top: representative passive responses to hyperpolarizing stimuli from neurons isolated from 12-, 64-, and 115-week-old mice. Blue traces correspond to 0 pA injection, red traces to –10 pA, and black traces to –40 pA. Bottom: voltage–current relationship of top recordings. (**G**) Comparison of the input resistance between different ages. Data points are from N = 4 animals, n = 35 cells, from 12 weeks old, N = 6 animals, n = 65 cells, from 64 weeks old, and N = 4 animals, n = 32 cells, from 115 weeks old. Error bars represent SEM. p-Values are shown at the top of the graphs. Red p-values indicate p-values<0.05, while black p-values indicate p-values>0.05.

higher percentage of old neurons firing spontaneously compared to middle-aged neurons, the mean firing frequency in old neurons (60 ± 20 action potentials [APs]/min) was significantly lower than that observed in middle-aged neurons (177 ± 41 APs/min; *Figure 3E*).

We measured the input resistance, another passive membrane property, in cells polarized to a potential of –65 mV by injecting a negative holding current that did not exceed –50 pA. Cells requiring larger holding currents were excluded. From the response to additional negative current pulses (–40 to 0 pA, *Figure 3F*), we calculated the input resistance (ΔV/injected current). At these membrane potentials, without activation of voltage-activated ion channels, the change in voltage is proportional to the injected current. Input resistance was not affected by age. Young adult neurons (0.8 ± 0.09 GΩ), middle-aged neurons (1.1 ± 0.06 GΩ), and old neurons (1.1 ± 0.11 GΩ) exhibited similar input resistance (*Figure 3G*). In conclusion, our findings suggest that aging leads to a gradual depolarization of the RMP without significant alteration of the input resistance.

## Sympathetic motor neurons from old mice are more responsive to electrical stimulation

During sympathetic reflexes, the peripheral motor neurons increase their firing frequency in response to commands from the central sympathetic nuclei. These commands can be mimicked as direct electrical stimulation, as shown in *Figure 4A*. To test whether aging increases the sensitivity of sympathetic motor neurons to electrical stimulation, neurons were injected with 1 s current steps of amplitude

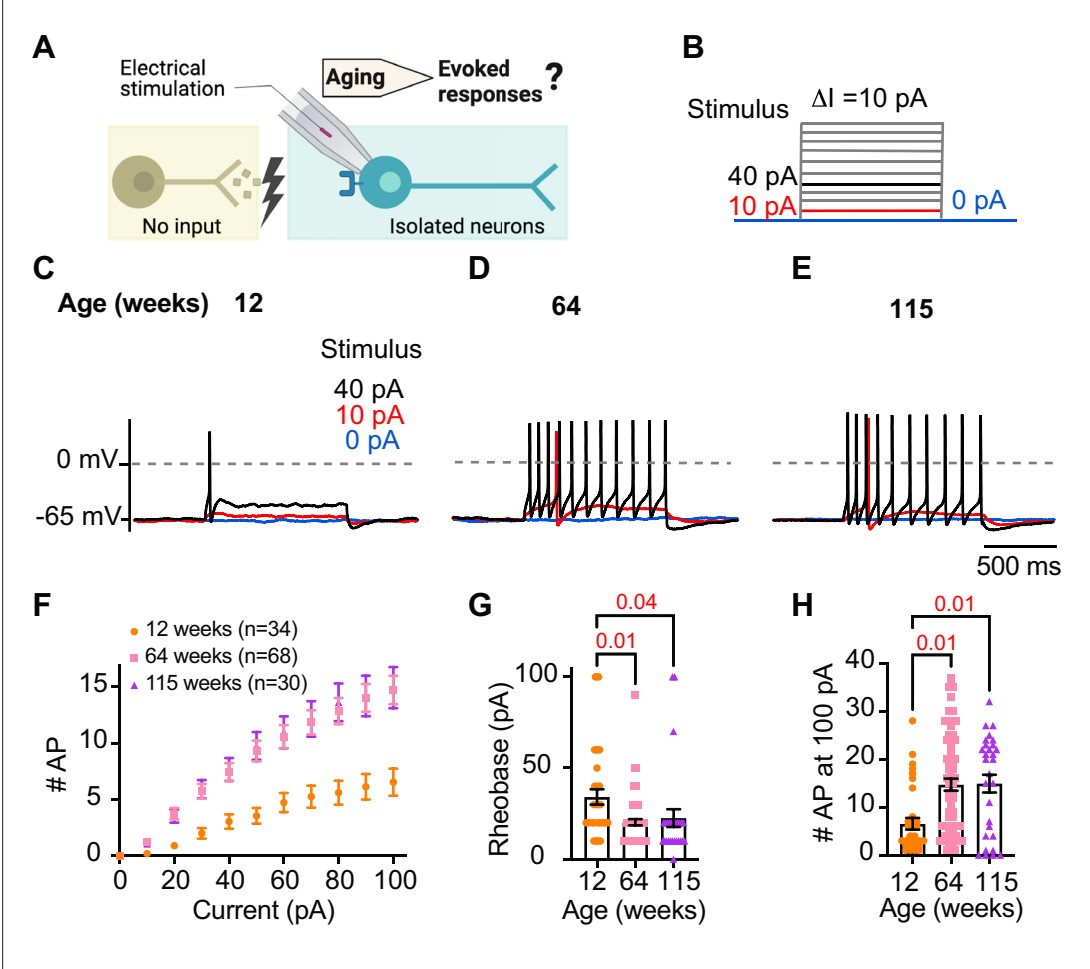

**Figure 4.** Sympathetic motor neurons from old mice are more responsive to electrical stimulation. (**A**) In the context of the sympathetic reflex, motor neurons increase their firing frequency in response to inputs from the sympathetic nuclei in the brain. In this study, motor neuron responses were measured using electrical stimulation at different ages. (**B**) Stimulation protocol to mimic preganglionic input. (**C–E**) Representative voltage responses of sympathetic motor neurons from 12- (**C**), 64- (**D**), and 115-week-old mice (**E**). Blue, red, and black traces are in response to 0, 10, and 40 pA current injection, respectively. The dotted line shows 0 mV as reference. (**F**). Comparison of stimulation–response curves of the number of action potentials (APs) vs. injected current between different ages. (**G**) Comparison of the minimum current injected that elicited at least one AP (Rheobase) between different ages. (**H**) Comparison of the number of APs fired at the maximum stimulus (100 pA) between different ages. Data were collected from N = 4 animals, n = 34 cells, from 12 weeks old, N = 6 animals, n = 68 cells, from 64 weeks old, and N = 4 animals, n = 30 cells, from 115 weeks old. Error bars represent SEM. p-Values are shown at the top of the graphs.

ranging from 10 pA to 100 pA starting from a potential of –65 mV ('0 pA current') (*Figure 4B*). Young neurons exhibited a characteristic firing pattern demonstrated in the left panel of *Figure 4C*. With injections of 10 pA, no AP was elicited (red trace), and with a 40 pA stimulus lasting 1 s, only one AP was evoked (black trace), illustrating the phenomenon known as spike adaptation that is typical of young sympathetic motor neurons. In contrast, both middle-aged and old neurons displayed a different response. A 10 pA stimulus was sufficient to elicit APs in middle-aged and old neurons (*Figure 4D and E*), and in middle-aged neurons, the 40 pA stimulus induced the firing of eleven APs in a typical example (*Figure 4D*), and similarly, in old neurons, it led to the firing of 10 APs (*Figure 4E*).

To assess the impact of age on the excitability of sympathetic motor neurons in a population sample, we compared stimulus–response curves, rheobase values (minimum current needed to elicit at least one AP), and the number of APs elicited with the maximum stimulus. Our results revealed that middle-aged and old neurons fired more APs with each stimulus (*Figure 4F*) than young neurons. Older neurons exhibited a reduced threshold for eliciting APs (64 weeks: 20 ± 2 pA; 115 weeks: 23 ± 5 pA) compared to young neurons (34 ± 4 pA, *Figure 4G*). Furthermore, older neurons fired more APs

with a 100 pA current injection (64 weeks: 15 ± 1 APs; 115 weeks: 15 ± 2 APs) compared to young neurons (12 weeks: 7 ± 1 APs, *Figure 4H*). These findings support the concept that aging leads to increased excitability of sympathetic motor neurons.

## Analysis of neuronal subpopulations

Sympathetic motor neurons display stereotyped distinct repetitive firing patterns, which classify them into three categories: tonic (class I), phasic (class II), and adapting (class III). (*Malin and Nerbonne, 2000*; *Malin and Nerbonne, 2001*; *Springer et al., 2015*; *Kim et al., 2019*). In our experiments, we classified cells based on their responses to a supra-threshold current injection (20 pA above their rheobase), ensuring that the firing response was not saturated. Representative traces of firing patterns from middle-aged neurons are illustrated in *Figure 5A*. The increased number of APs at maximal stimulus compared with 20 pA above the rheobase shows that the firing response was not saturated during classification (*Figure 5B*). In agreement with previous reports, subpopulations showed differences in frequency–stimulus curves (*Figure 5C*). Also, tonic cells have a more depolarized RMP (*Figure 5D*, left, tonic = –53 ± 2 mV, phasic = –61 ± 1 mV, adapting = –60 ± 2 mV), while adapting neurons have a lower input resistance (*Figure 5D*, right, tonic = 1.13 ± 0.08 GΩ, phasic = 1.16 ± 0.09 GΩ, adapting = 0.55 ± 0.18 GΩ). Tonic cells were also more spontaneously active (*Figure 5D*, middle). In general, the responses from the three neuronal subpopulations are consistent with previous reports.

Next, we examined the effect of aging on neuronal subpopulations. We observed an altered subtype distribution in older neurons (*Figure 5E*, p-value<0.0001 using a contingency table and chi-square/Fisher test). The percentage of adapting neurons decreased in middle age (10%) and old (12%) compared to young (29%). The percentage of phasic neurons decreased only in old (42%) compared to middle-aged (56%) and young (55%) neurons. Accordingly, the percentage of tonic neurons increased in middle-aged (34%) and old (46%) compared to young (19%) neurons. These findings highlight how aging impacts the distribution of firing subtypes in sympathetic motor neurons.

While age does impact the distribution of firing subtypes, the question remains whether intrinsic properties are affected within each subpopulation across different ages. Therefore, we analyzed the intrinsic properties across subpopulations between different age groups. The RMP of adapting and phasic firing neurons was more depolarized in middle-aged and old compared to young neurons (*Figure 5F*). The RMP of tonic neurons tended to become more depolarized with age, but the effect was not significant (*Figure 5F*). Similar to the observation when the entire population was analyzed, the input resistance was not significantly different with age within phasic and adapting neurons. The input resistance of tonic neurons was significantly larger only in old (1.44 ± 0.16 GΩ) neurons compared to young (0.93 ± 0.16 GΩ) neurons (*Figure 5G*). This analysis suggested that, regardless of the subpopulation type, older neurons tend to have more depolarized RMP with no changes in input resistance.

Interestingly, aging leads to an increase in the number of spontaneous APs in phasic and tonic neurons (*Figure 5H*). In the phasic class, middle-aged neurons fired 3 ± 2 APs per minute and old neurons fired 5 ± 3 APs compared to zero APs in phasic young neurons. In the tonic class, middle-aged neurons fired 179 ± 46 APs and old neurons fired 86 ± 27 APs, in striking contrast to zero APs in tonic young neurons. Next, we compared the maximum number of APs elicited by the strongest stimulus (100 pA, *Figure 5I*). In phasic neurons, the number of APs at 100 pA increased by 53% in middle age (9.5 ± 1.0 APs) and 133% in old (14.5 ± 2.1 APs) compared to young neurons (6.2 ± 1.0 APs). In tonic firing neurons, the number of APs at 100 pA increased by 36% in middle-aged (25 ± 1.4 APs) and 27% in old (24.1 ± 1.0 AP) compared to young neurons (19.0 ± 1.9 APs).

We conclude that the changes observed when categorizing neuronal classes occur across the entire population. In addition, aging is associated with a shift in the proportion of neurons falling in each class. As a result, we hypothesized that the underlying age-related molecular mechanism is broadly shared among sympathetic motor neurons and plays a role in controlling the firing frequency.

## Aging reduces KCNQ current

We next directed our attention to identifying molecular candidates underlying changes in membrane excitability. Sympathetic motor neurons express at least one voltage-gated sodium channel isoform ($Na_V1.7$) (*Toledo-Aral et al., 1997*; *Schofield et al., 2008*) and several voltage-gated potassium channels (*Dixon and McKinnon, 1996*; *Shi et al., 1997*). Specifically, $K_V4$, $K_V2$, and $K_V7$ (KCNQ) channels are crucial in controlling the RMP, rheobase, firing frequency, and spike adaptation (*Malin*

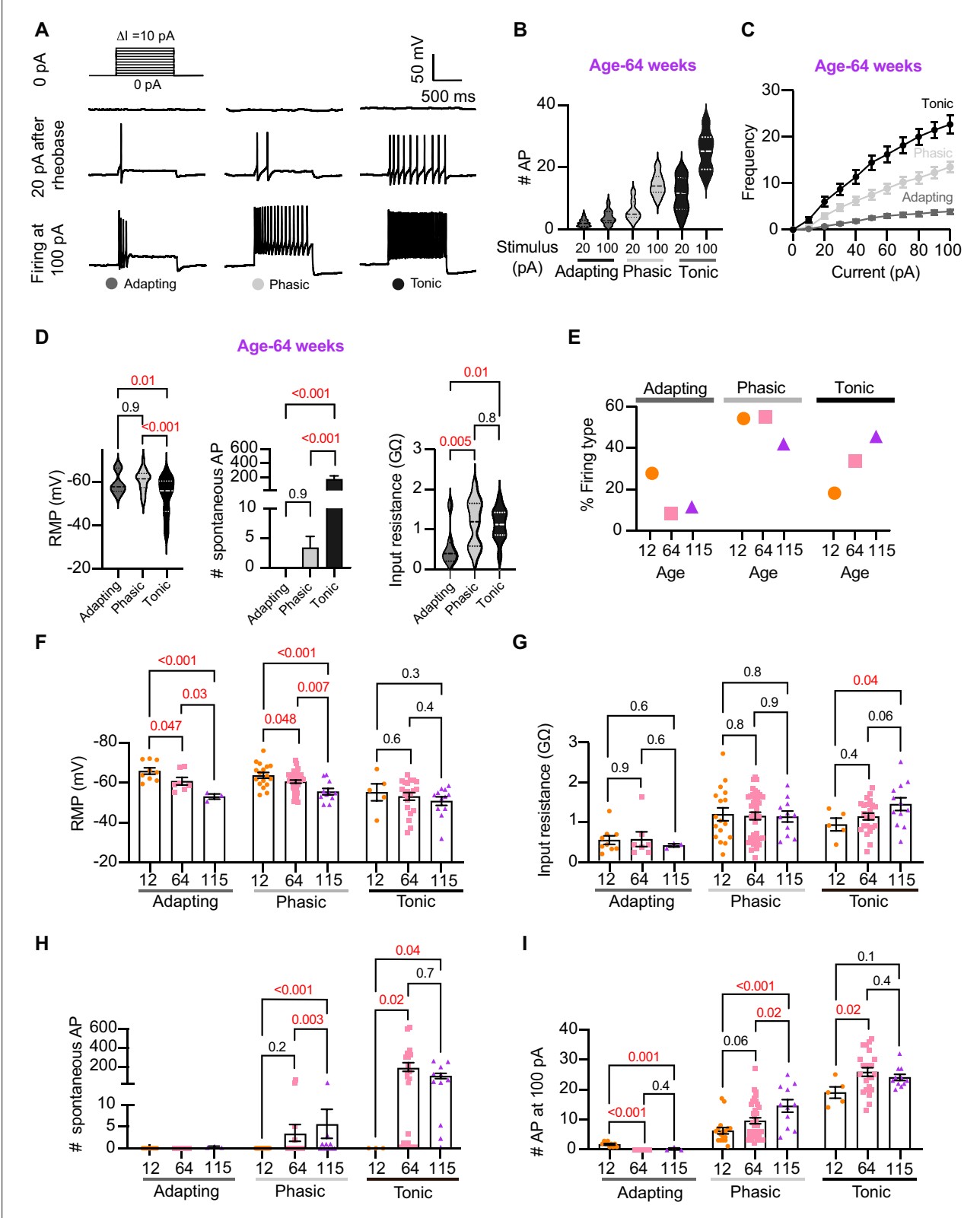

**Figure 5.** Analysis of neuronal subpopulations. Aging shifts neuronal population toward tonic firing. (**A**) Representative recordings from three different neurons illustrate the variability in the response to show the method used to classify cells by their firing pattern as adapting (left), phasic (center), or tonic (right). Responses to stimulation of 0 (top), 20 pA above the rheobase (middle), and 100 pA (bottom). The classification was based on the response 20 pA above the rheobase. (**B**) Comparison of the number of action potentials (APs) elicited by current injections of 20 pA more than rheobase or

*Figure 5 continued on next page*

*Figure 5 continued*

100 pA between classes. Data are from 64-week-old mice. (**C**) Comparison of the stimulus–frequency curves between classes. Data are from 64-week-old mice. (**D**) Comparison of RMP, (left), number of spontaneous APs (middle), and input resistance (right) between classes. All data are from 64-week-old mice. (**E**) Comparison of the percentage of neuronal firing subtypes between different ages. (**F**) Comparison of the RMP between ages and divided into neuronal subpopulations. (**G**) Comparison of the input resistance between ages and divided into neuronal subpopulations. (**H**) Comparison of the number of spontaneous APs between ages and neuronal subpopulations. (**I**) Comparison of the number of APs in response to maximum stimulation (100 pA) between ages and neuronal subpopulations. Data points for the adapting subpopulation are nine cells from 12 weeks old, seven from 64 weeks old, and three from 115 weeks old. Data points for the phasic subpopulation are 17 cells from 12 weeks old, 38 from 64 weeks old, and 11 cells from 115 weeks old. Data points for the tonic subpopulation are 5 cells from 12 weeks old, 23 from 64 weeks old, and 11 cells from 115 weeks old. Data points are also from a total of four 12-week-old mice, six 64-week-old mice, and four 115-week-old mice. Error bars represent SEM. Red values indicate p-values<0.05, while black values indicate p-values>0.05.

*and Nerbonne, 2000*; *Malin and Nerbonne, 2001*; *Liu and Bean, 2014*). These channels are of particular interest as potential contributors to the age-related alterations in neuronal excitability that we observed.

Previous work by our group and others demonstrated that cholinergic stimulation leads to a decrease in M current and increases the excitability of sympathetic motor neurons at young ages (*Brown and Adams, 1980*; *Brown et al., 1981*; *Brown and Passmore, 2009*; *Lamas et al., 2002*; *Miranda et al., 2013*; *Shapiro et al., 2000*; *Vivas et al., 2014*; *Wang and McKinnon, 1995*; *Zaika et al., 2006*). The molecular determinants of the M current are channels formed by KCNQ2 and KCNQ3 in these neurons (*Shapiro et al., 2000*; *Wang et al., 1998*; *Selyanko et al., 2000*). Thus, *Figure 6A* shows a voltage response (measured in current-clamp mode) and a consecutive M current recording (measured in voltage-clamp mode) in the same neuron upon stimulation of Gq-coupled M1 mAChRs. It illustrates the temporal correlation between the decrease in M current with the increase in excitability and firing of APs. This strong dependence led us to hypothesize that aging decreases M current, leading to a depolarized RMP and hyperexcitability (*Figure 6B*). For these experiments, we measured the RMP and evoked activity using perforated patch, followed by the amplitude of M current using a whole-cell voltage clamp in the same cell. We also measured the membrane capacitance as a proxy for cell size. Interestingly, M current density was smaller by 29% in middle-aged (7.5 ± 0.7 pA/pF) and by 55% in old (4.8 ± 0.7 pA/pF) compared to young (10.6 ± 1.5 pA/pF) neurons (*Figure 6C and D*). The average capacitance was similar in young (30.8 ± 2.2 pF), middle-aged (27.4 ± 1.2 pF), and old (28.8 ± 2.3 pF) neurons (*Figure 6E*), suggesting that aging is not associated with changes in cell size of sympathetic motor neurons, and supporting the hypothesis that aging alters the levels of M current. Next, we tested the effect on the abundance of the channels mediating M current. Contrary to our expectation, we observed that KCNQ2 protein levels were 1.5 ± 0.1-fold higher in old compared to young neurons (*Figure 6F and G*). Unfortunately, we did not find an antibody to detect consistently KCNQ3 channels. We concluded that the decrease in M current is not caused by a decrease in the abundance of KCNQ2 protein.

To explore the hypothesis that a reduction in M current is responsible for the age-associated depolarization of the RMP, we compared these two parameters measured in the same cells. We observed a correlation between M current and RMP in young (coefficient of determination ($r^2$) = 0.22, p-value for the correlation fit = 0.007, *Figure 6H and K*) and middle-aged neurons ($r^2$ = 0.20, p-value for the correlation fit = 0.002, *Figure 6I and K*). In old neurons, the M current and RMP were no longer correlated ($r^2$ = 0.05, p-value for the correlation fit = 0.1, *Figure 6J and K*). *Figure 6K* shows the decrease in the coefficient of determination ($r^2$) with aging. Similarly, the M current amplitude also correlated well with the number of APs elicited at 100 pA in young and middle-aged neurons but not in old ones. The variance in M current amplitude explained 32% of the variation in the number of APs in young ($r^2$ = 0.32, p-value for the correlation fit = 0.001, *Figure 6L*) and 24% in middle-aged neurons ($r^2$ = 0.24, p-value for the correlation fit = 0.0001, *Figure 6M*). In old neurons, the variance in M current amplitude explained only 0.05% of the variation in the number of APs ($r^2$ = 0.05, p-value for the correlation fit = 0.15, *Figure 6N*). *Figure 6O* shows the decrease in the coefficient of determination with aging for the number of APs. These analyses support the hypothesis that a reduction in M current alters the electrical properties, and in the case of old neurons, the marked decrease in M current compromises its role in maintaining the RMP and spontaneous firing.

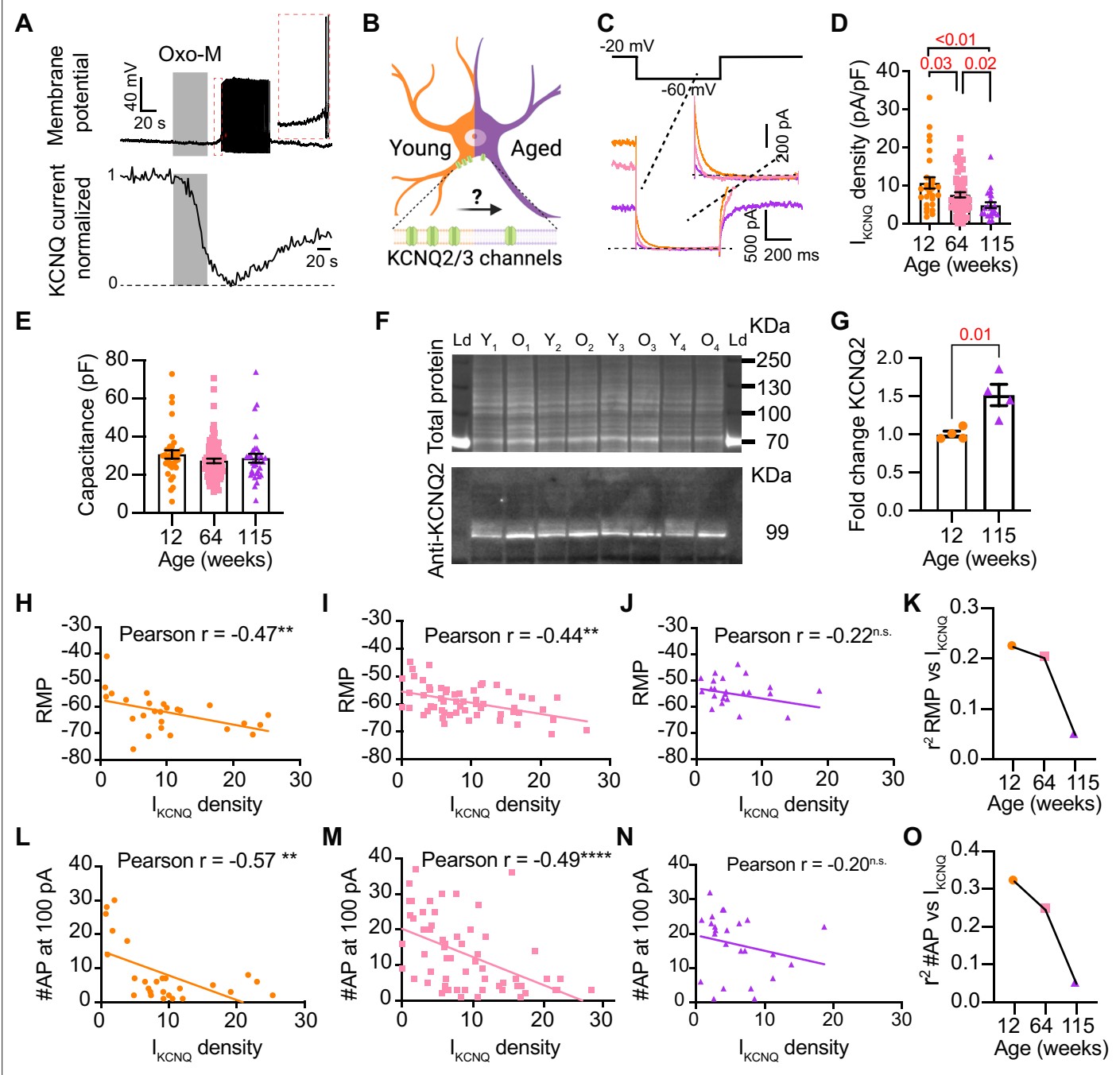

**Figure 6.** Sympathetic motor neurons from old mice show a KCNQ current reduction. (**A**) Recordings illustrating relevance of KCNQ current for controlling membrane potential and firing in sympathetic motor neurons. Top: voltage response to inhibition of KCNQ channels with 10 µM Oxo-M; bottom: normalized KCNQ current recording in response to oxo-M in the same cell after going whole cell. (**B**) Schematic representation of the hypothesis that aged cells show a reduced activity of KCNQ channels. (**C**) KCNQ current recordings from neurons isolated from mice of different ages (orange 12 weeks old, pink 64 weeks old, and purple 115 weeks old) in response to a voltage step (top). Inset shows an expanded view of the current tail. (**D**) Comparison of KCNQ current density between different ages. Data points are from N = 5 animals, n = 27 cells, from 12 weeks old, N = 8 animals, n = 62 cells, from 64 weeks old, and N = 5 animals, n = 24 cells, from 115 weeks old. (**E**) Comparison of capacitance between ages. (**F**) Blot stained for total and KCNQ2 protein collected from superior cervical ganglia from 12- and 115-week-old mice. (**G**) Comparison of fold change of KCNQ2 abundance relative to total protein between ages (n = 4 blots from different mice). (**H–J**) Linear correlation between RMP and KCNQ current density in 12-week-old (**H**), 64-week-old (**I**), and 115-week-old mice (**J**). (**K**) Comparison of the determination coefficient for the RMP and KCNQ current between ages. (**L–N**) Linear correlation between the number of action potentials (APs) at 100 pA and KCNQ current density in 12-week-old (**L**), 64-week-old (**M**), and 115-week-old mice (**N**). (**O**) Comparison of the determination coefficient for the maximum #AP and KCNQ current between ages. Data

*Figure 6 continued on next page*

*Figure 6 continued*

points are from N = 4 animals, n = 26 cells, from 12 weeks old, N = 6 animals, n = 58 cells, from 64 weeks old, and N = 4 animals, n = 24 cells, from 115 weeks old. Error bars represent SEM.

The online version of this article includes the following source data for figure 6:

**Source data 1.** PDF file containing original western blots for *Figure 6F*, indicating the relevant bands and comparisons.

**Source data 2.** Original files for western blot analysis displayed in *Figure 6F*.

## Other voltage-gated sodium and potassium currents are not altered with aging

Loss of voltage-gated potassium channel function, including the current of $K_V2$ and $K_V4$, has also been invoked in aging and age-associated memory decline (*Fenyves et al., 2021*; *Frazzini et al., 2016*; *Navarro-García et al., 2022*; *Sesti, 2016*; *Simkin et al., 2015*; *Yu et al., 2019*). Therefore, we also looked for potential age-associated changes in other voltage-gated potassium currents. We used a recording solution designed to abolish sodium, calcium, and potassium currents mediated by both calcium-activated potassium channels (BK channels) and KCNQ channels. This recording solution contained 100 nM tetrodotoxin (TTX) to suppress voltage-gated sodium channels, 100 μM $Cd^{2+}$ to inhibit calcium channels and 10 μM XE-991 to block KCNQ channels. Next, we compared the outward current density before and after application of a cocktail of 100 nM phiroxotoxin and 100 nM guangxi-toxin to block $K_V4$ and $K_V2$ channels. *Figure 7A and C* shows representative traces and the comparison of the current density between young and old neurons. We did not observe significant differences in the potassium current insensitive to XE-991 between groups (young = 363 ± 20 pA/pF, old = 342 ± 30 pA/pF, *Figure 7A and B*) nor in the $K_V2$ and $K_V4$-sensitive currents (young = 157 ± 31 pA/pF, old = 110 ± 17 pA/pF, *Figure 7C and D*).

In nociceptive neurons, similar to sympathetic neurons, modulation of $Na_V1.7$ channels has been linked to increased excitability (*Akin et al., 2021*; *Dib-Hajj et al., 2013*; *Li et al., 2018*). Additionally, exogenous expression of $Na_V1.7$ channels in non-excitable cells induces cell senescence (*Warnier et al., 2018*), further highlighting a significant connection between aging and voltage-gated sodium channels. To investigate the hypothesis that aging might contribute to hyperexcitability through alterations in sodium currents in sympathetic motor neurons, we conducted experiments using 5 ms voltage steps in a low-sodium recording solution. We found that the sodium current density was similar in young (58.9 ± 5.4 pA/pF), middle-aged (59.0 ± 5.3 pA/pF), and old (50.1 ± 5.7 pA/pF) neurons (*Figure 7E–G*). In conclusion, our data indicate that aging does not alter the $Na_V$ and $K_V$ currents insensitive to XE-991 in sympathetic motor neurons.

## Pharmacological inhibition of KCNQ channels mimics aged phenotype while pharmacological activation of KCNQ channels mimics young phenotype

To further test the hypothesis that the decrease in M current is responsible for the hyperexcited state in old neurons, we used a pharmacological approach targeting KCNQ channels and assessed whether the age-dependent phenotype could be mimicked or reversed. To inhibit KCNQ channels, we used linopirdine at a concentration of 25 μM. Linopirdine inhibits both KCNQ2 and KCNQ3 channels with an $IC_{50}$ of around 5 μM (*Alexander et al., 2019*), with an inhibition time constant of around 3 s at negative potentials (*Greene et al., 2017*). In the presence of linopirdine, the RMP of young neurons became less negative, from –64.3 ± 5.1 mV to –50.4 ± 5.1 mV on average. In five out of eight cases, neurons that were not firing started firing after washing linopirdine into the bath (*Figure 8A and B*).

Conversely, we used retigabine at a concentration of 10 μM to increase M current. This is achieved as retigabine induces both a negative shift in the voltage dependence of activation and a voltage-independent increase in the open probability of KCNQ2 and KCNQ3 channels. Retigabine activates KCNQ2 with an $IC_{50}$ of 2.5 μM and KCNQ3 with an $IC_{50}$ of 600 nM (*Alexander et al., 2019*). The onset of its action takes about 3 min, according to information provided by the manufacturer. In the presence of retigabine, the RMP of old neurons became more negative, from –49.9 ± 3.8 mV to –74.7 ± 3.0 mV on average. Aged neurons that were firing APs stopped firing after the activation of KCNQ channels, and some of these fired again after washing out the drug (*Figure 8C and D*). In fact, the RMP became more positive (–58.2 ± 2 mV) after removal of retigabine. Together, these findings

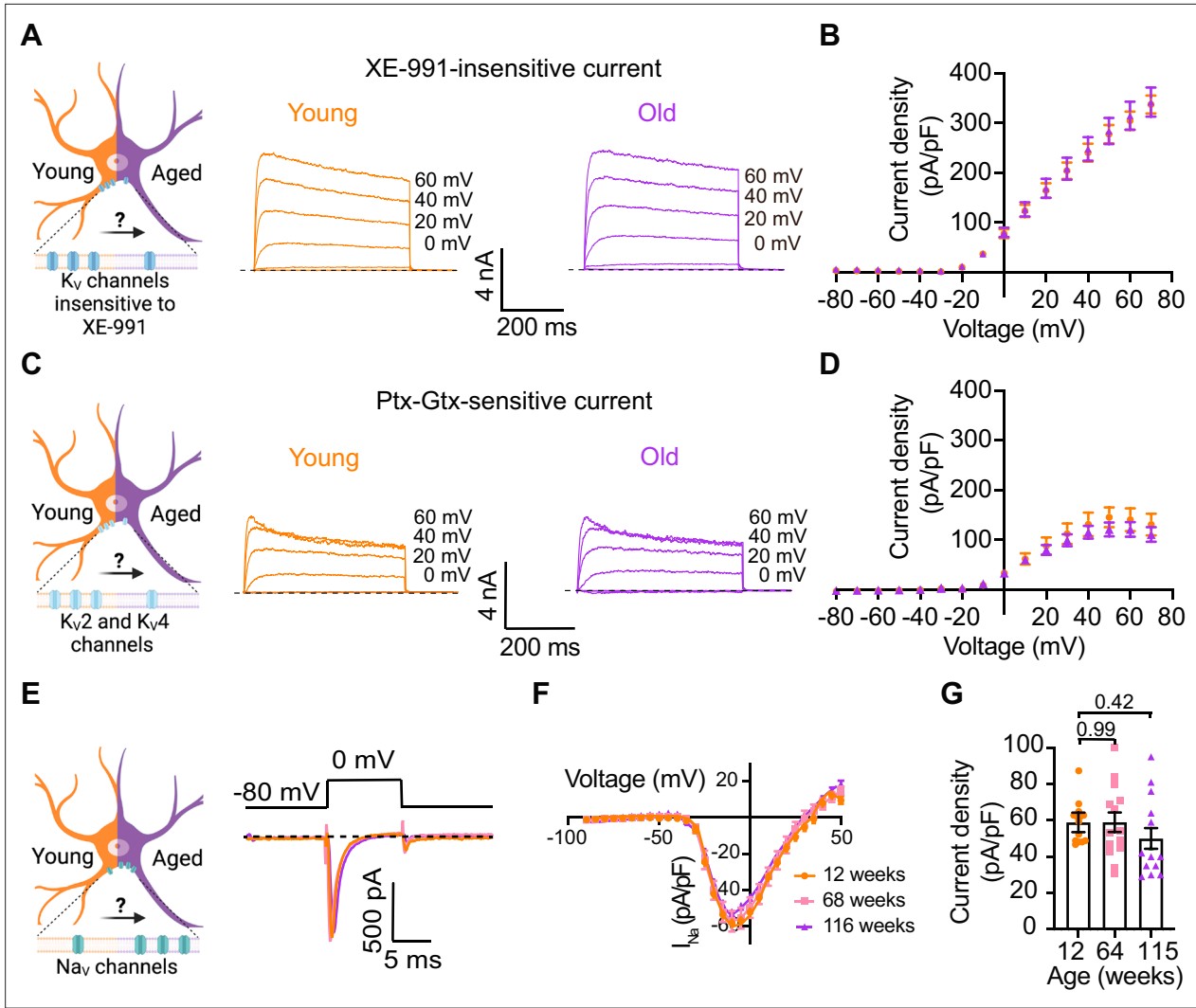

**Figure 7.** Sympathetic motor neurons from old mice did not show changes in sodium currents or XE-991-insensitive potassium currents. (**A**) Left: schematic of the hypothesis that $K_V$ channels insensitive to XE-991 are reduced in aged cells. Right: family of outward currents in the presence of KCNQ channel blocker, 100 nM XE-991, in 12-week-old and 115-week-old mice. Text at the right of each current trace corresponds to the voltage used to elicit that current. (**B**) Current–voltage relationship of the steady-state $K_V$ currents insensitive to XE-991 from cells isolated from 12-week-old and 115-week-old mice. Data points of $K_V$ currents are from N = 3 animals, n = 9 cells, from 12 weeks old, and N = 3 animals, n = 8 cells, from 115 weeks old. (**C**) Left: schematic of the hypothesis that $K_V2$ (Guangxitoxin-1E sensitive, Gtx) and $K_V4$ (phrixotoxin-1 sensitive, Ptx) channels are reduced in aged cells. Right: family of Ptx- and Gtx-sensitive outward currents in 12-week-old and 115-week-old mice. Text at the right of each current trace corresponds to the voltage used to elicit that current. (**D**) Current–voltage relationship of the steady-state $K_V$ currents sensitive to Ptx- and Gtx from cells isolated from 12-week-old and 115-week-old mice. Data points of $K_V2$ and $K_V4$ currents are from N = 3 animals, n = 12 cells, from 12 weeks old, and N = 3 animals, n = 8 cells, from 115 weeks old. (**E**) Left: schematic of the hypothesis that $Na_V$ channels sensitive to TTX are increased in aged cells. Right: representative sodium current recordings from neurons isolated from mice of different ages (orange, 12 weeks old; pink, 64 weeks old; and purple, 115 weeks old) in response to a voltage step (top). (**F**) Current–voltage relationship of the peak sodium current for different ages. (**G**) Comparison of sodium current density between different ages. Data points of $Na_V$ currents are from N = 3 animals, n = 12 cells, from 12 weeks old, N = 3 animals, n = 14 cells, from 64 weeks old, and N = 3 animals, n = 13 cells, from 115 weeks old. Error bars represent SEM. p-Values are shown at the top of the graphs.

support the hypothesis that a reduction of KCNQ channel activity is responsible for the hyperexcitability of sympathetic motor neurons in aged animals.

## Discussion

This research investigates the cellular and molecular mechanisms underlying age-associated sympathetic overactivity. Our results support the idea that, alongside age-related central changes, the

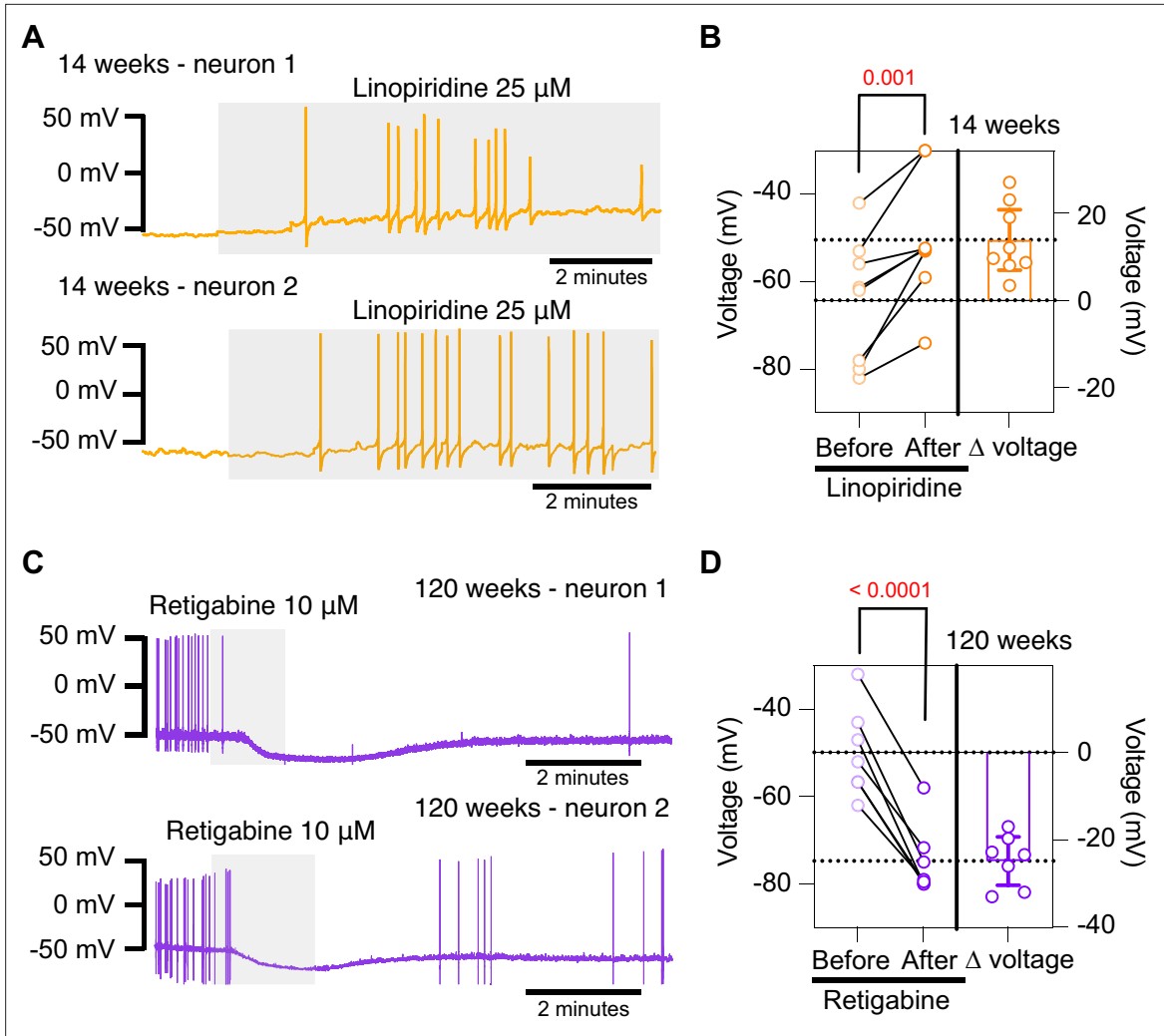

**Figure 8.** Pharmacological inhibition and activation of KCNQ channels mimic the age-dependent phenotype. (**A**) Membrane potential recordings from two young neurons treated with 25 µM linopirdine during the time illustrated by the light gray box. No holding current was applied. (**B**) Left: summary of the resting membrane potential measured before (light orange) and after (dark orange) the application of linopiridine. Right: summary of the depolarization produced by linopirdine calculated by subtracting the post-drug voltage from the pre-drug voltage (ΔV). Data points are from N = 2 animals, n = 8 cells, 14-week-old mice. (**C**) Membrane potential recordings from two aged neurons treated with 10 µM retigabine during the time illustrated by the light gray box. No holding current was applied. (**D**) Left: summary of the resting membrane potential measured before (light purple) and after (dark purple) the application of retigabine. Right: summary of the hyperpolarization produced by retigabine calculated by subtracting the post-drug voltage from the pre-drug voltage (ΔV). Data points are from N = 2 animals, n = 7 cells, 120-week-old mice. Error bars represent 95% CI. p-Values are shown at the top of the graphs.

peripheral component of the sympathetic autonomic reflex is also affected by aging. Key findings are that aging influences the intrinsic membrane properties of sympathetic motor neurons in the following ways: (1) older motor neurons exhibit a more positive RMP and increased rheobase, (2) the percentage of motor neurons displaying spontaneous activity increases with age, (3) older motor neurons respond with higher firing rates to electrical stimulation, (4) older motor neurons show a predominant tonic firing subpopulation, (5) older neurons exhibit reduced M current, and (6) spontaneous activity in aged neurons can be reversed using pharmacological activation of KCNQ2/3 channels.

## Age-related changes in neuronal excitability

The decline in nervous system function during healthy aging has been attributed to alterations in the intrinsic membrane properties of neurons and glial cells (*Table 1*). These changes in electrical behavior have been observed in various experimental models, ranging from simple organisms like

**Table 1.** Age-related changes in neuronal excitability.

The table compares the changes in intrinsic properties and the underlying mechanisms. It also lists cell types, research models, and references.

| Excitability | Intrinsic properties | Ion channel/mechanisms | Cell type | Research model | Reference |
|---|---|---|---|---|---|
| Decrease | RMP-unaffected Input resistance-unaffected Rheobase-increase | Decrease in cholinergic response | CA1 pyramidal neurons slices | Sprague– Dawley rats, 3–4 vs 25–32 months old | *Potier et al., 1992* |
| Decrease | RMP-unaffected Input resistance-unaffected Rheobase-increase | - | Neostriatal neurons slices | Fischer 344 rats, 3–5 vs 24–26 months old | *Cepeda et al., 1992* |
| - | RMP-unaffected Input resistance-unaffected Rheobase-unaffected | Decrease in cholinergic response | CA1, CA3 and fascia dentata | F-344 rats, 3 weeks vs 9 months vs 24–27 months old | *Shen and Barnes, 1996* |
| - | - | Increase in L-type Ca$_v$ channel activity | CA1 hippocampal neurons Zipper slices | F-344 rats, 3–6 vs 12–14 vs 23–26 months old | *Thibault and Landfield, 1996* |
| - | - | Decrease in NMDA protein levels | CA1 and CA3 hippocampus | Sprague–Dawley rats 3–9 vs 12–17 vs 18–24 vs 25–28 vs 29–31 months old | *Wenk and Barnes, 2000* |
| Unaffected | RMP-unaffected Input resistance-increase | - | Dentate granule cells slices | Rhesus monkeys, 11 vs 24 years old | *Luebke and Rosene, 2003* |
| Increase | RMP-unaffected Input resistance-increase Rheobase-decrease | - | Layer 2/3 pyramidal neurons/ prefrontal cortex | Rhesus monkeys, 8 vs 22 years old | *Chang et al., 2005* |
| Decrease | - | Increase in cAMP signaling and KCNQ/HCN activity | Dorsolateral prefrontal cortical/DELAY neuron slices | Rhesus monkeys, 7–9 vs 12–13 vs 17–21 years old | *Wang et al., 2011* |
| - | - | Decrease in KCNQ expression | RT-PCR-whole brain | *Drosophila*, 5, 25, 40, 50, and 60 days of age | *Cavaliere et al., 2013* |
| Increase | RMP-unaffected Input resistance-increase | | L3 pyramidal neurons/ lateral prefrontal cortex slices | Rhesus monkeys, 8 vs 22 years old | *Coskren et al., 2015* |
| - | RMP-more negative Input resistance-decreases | Increase in inward and outward rectifier K$^+$ currents | Microglia in striatum, neocortex and entorhinal cortex slices | C57BL6 mice, 2–3 vs 19–24 months old | *Schilling and Eder, 2015* |
| Increase | RMP-unaffected Input resistance-unaffected | Increase in K$_v$4.2/K$_v$4.3 expression | CA3 hippocampal neurons Slices | F1 hybrid Fischer 344x Brown Norway rats, 2–5 vs 29–32 months old | *Simkin et al., 2015* |
| Increase | - | Decrease in BK current | Dorsal root ganglion neuron | Wistar male rats, 3 and 18 months | *Yu et al., 2015* |
| Increase | - | Increase in glutamate-gated chloride and L-type calcium channels function | | *Caenorhabditis elegans* | *Zullo et al., 2019* |
| Increase | RMP-unaffected Rheobase-decreases | Increase in I$_h$ | L5 pyramidal neurons Somatosensory cortex slices | Mice C57BL/6, 2–6 vs 18–29 months old | *Popescu et al., 2021* |
| Increase | RMP-unaffected Input resistance-increase Rheobase-decrease | - | CA3 hippocampal interneurons | Sprague–Dawley, 5 vs 24 months old | *Griego and Galván, 2022* |

*Table 1 continued on next page*

de La Cruz *et al.* eLife 2023;12:RP91663. DOI: https://doi.org/10.7554/eLife.91663

*Table 1 continued*

| Excitability | Intrinsic properties | Ion channel/mechanisms | Cell type | Research model | Reference |
|---|---|---|---|---|---|
| Increase | RMP-more positive<br>Input resistance-unaffected<br>Rheobase-unaffected | Decrease in KCNQ2/3 current | hypocretin/orexin neurons Hypothalamus | C57BL/6J,<br>2–3 vs 18 months old | *Li et al., 2022* |
| Increase | RMP-more positive<br>Input resistance-increases<br>Rheobase-decreases | Increase in glial–neuronal coupling | Dorsal root ganglion | Balb/c mice,<br>3 vs 12 vs 17 months old | *Hanani et al., 2023* |

*Caenorhabditis elegans* and *Drosophila* to more complex ones like rodents and monkeys. Notably, age-related hyperexcitability has emerged as a predominant characteristic in neurons across different brain regions (*Table 1*).

Research on this topic has emphasized the central nervous system. However, recent evidence suggests that aging also impacts the intrinsic properties of peripheral sensory neurons leading to hyperexcitability (*Hanani et al., 2023*). In line with this observation, our research reveals increased excitability in old sympathetic motor neurons. Collectively, these findings underscore that aging affects the intrinsic properties of peripheral neurons, challenging the notion that age-related changes are limited to the central nervous system. Hence, it is crucial to reconsider our understanding of aging-related sympathetic overactivity with a holistic perspective. Investigating how aging affects the intrinsic properties of sensory, central, and sympathetic motor neurons throughout a lifetime will be essential research to comprehend the underlying cause-and-effect mechanisms.

## The molecular mechanism underlying the age-related excitability changes

The expression, distribution, regulation, and function of ion channels play a pivotal role in determining the intrinsic membrane properties of neurons. The mechanism underlying changes in age-associated excitability encompass alterations in cholinergic and glutamatergic responses and the function and expression of voltage-gated ion channels, such as $Ca_V$, $K_V$, and HCN. In recent years, research on aging has revealed a significant impact of ion channels in the aging process. For example, repressing glutamate-gated chloride channels and L-type channels extends longevity in *C. elegans* (*Zullo et al., 2019*). Furthermore, inhibiting A-type $K^+$ channels has shown the potential to revert the intrinsic excitability of aged CA3 pyramidal neurons to a young-like state in rats (*Simkin et al., 2015*).

Our data show a decrease in M current and suggest that it is a key mechanism behind the development of a more depolarized RMP and hyperexcitability in old sympathetic motor neurons. Work done in other systems has also underscored the importance of M current in the development of age-associated neuronal dysfunction. In *Drosophila*, aging is associated with reduced M expression in the brain, while KCNQ overexpression in mushroom body neurons reverses age-related memory impairment (*Cavaliere et al., 2013*). Similarly, in macaques, blocking KCNQ channels partially restores memory-related firing of aged neurons to more youthful levels (*Wang et al., 2011*). Recent findings by Li et al. demonstrate that aged hypocretin/orexin neurons exhibit hyperexcitability with lower KCNQ expression. Their study shows that selectively disrupting *Kcnq2/3* genes in young hypocretin neurons is sufficient to depolarize these neurons and cause sleep fragmentation, mimicking the sleep instability observed in aged mice (*Li et al., 2022*). In our work, we also analyzed the effect of aging on other currents without seeing significant differences, suggesting that KCNQ channels are particularly susceptible to a process occurring during aging.

In our study, we found a reduction in M current but an increase in KCNQ2 channel abundance; why? We speculate the increase in KCNQ2 abundance is a compensatory mechanism that does not achieve to restore the function of the cell. This experiment assessed total protein abundance, which includes synthesized protein in traffic and inserted at the plasma membrane. Our experiments did not investigate the abundance of KCNQ2/3 channels only in the plasma membrane. Multiple mechanisms, independent from increased synthesis, could underlie a reduction in M current including alterations in

traffic, insertion, post-translational modifications, and cofactors of KCNQ2 or KCNQ3 channels. This mechanism remains elusive and open for exploration.

## Limitations and conclusion

We want to point out that linopirdine has been reported to affect other ionic currents besides M current (*Lamas et al., 1997*; *Neacsu and Babes, 2010*). Despite this limitation, the application of linopirdine to young sympathetic motor neurons led to depolarization and firing of APs.

The hypothesis of age-associated hyperexcitability of sympathetic motor neurons should be tested in the neurons of other sympathetic ganglia, including the stellate and celiac. While our study allowed us to determine the effect of aging on sympathetic neurons in isolation from other components of the sympathetic reflex, it did not provide insight into whether the hyperexcitability of motor neurons is a compensatory mechanism responding to earlier changes in the brain or target tissues. This question has posed a long-standing challenge, especially in various pathologies where sympathetic over-activity, such as arrhythmias and hypertension, plays a role. In such cases, sympathetic overactivity could result from deterioration in the target organs or sensory components, such as the heart and carotid body. Further research and exploration are needed to unravel these complex interactions and establish a comprehensive understanding of the mechanisms underlying age-related changes in the sympathetic nervous system.

In conclusion, this study demonstrates that aging directly impacts the intrinsic electrical properties of sympathetic motor neurons. Furthermore, our research postulates that the decrease in KCNQ function underlies the hyperexcitability of sympathetic motor neurons. These findings shed light on the mechanisms involved in age-related changes within the sympathetic nervous system and offer a promising avenue for further investigation and potential intervention.

# Materials and methods

## Key resources table

| Reagent type (species) or resource | Designation | Source or reference | Identifiers | Additional information |
|---|---|---|---|---|
| Strain, strain background (*Mus musculus*, C57BL/6, males) | Mouse, wild type, young adult | Jackson Laboratory | RRID:IMSR_JAX:000664 | |
| Strain, strain background (*M. musculus*, C57BL/6, males) | Mouse, wild type, middle and old ages | Jackson Laboratory | RRID:IMSR_JAX:000664 | |
| Antibody | Anti-KCNQ2 (rabbit polyclonal) | ABCAM | Cat# ab22897; RRID:AB_775890 | 1:500 |
| Antibody | IgG (H+L)-HRP (goat anti-rabbit polyclonal) | Bio-Rad | Cat# 1706515; RRID:AB_11125142 | 1:15,000 |
| Chemical compound, drug | XE-991 | Alomone Labs | Cat# X-101 | |
| Chemical compound, drug | Phrixotoxin-1 | Alomone Labs | Cat# STP-700 | |
| Chemical compound, drug | Guangtoxin-1E | Alomone Labs | Cat# STG-200 | |
| Chemical compound, drug | Retigabine dihydrochloride | Alomone Labs | Cat# D-23129 | |
| Chemical compound, drug | Tetrodotoxin | Alomone Labs | Cat# T-550 | |
| Chemical compound, drug | Linopirdine | Sigma | Cat# L-134 | |
| Software, algorithm | ImageJ | ImageJ | RRID:SCR_003070 | |
| Software, algorithm | IGOR Pro | WaveMetrics | RRID:SCR_000325 | |
| Software, algorithm | Prism | GraphPad | RRID:SCR_002798 | |
| Software, algorithm | Excel | Microsoft | RRID:SCR_016137 | |

## Animal models

Male C57BL/6 WT mice were purchased from the Jackson Laboratory (12 weeks, RRID:IMSR_JAX:000664) or obtained from the NIA-NIH colony (ages 64 weeks and 115 weeks). All animals were kept in an animal facility with controlled conditions and were given standard chow and water ad libitum.

The animal handling protocol was approved by the University of Washington Institutional Animal Care and Use Committee.

## Sympathetic motor neuron cell culture

Neurons from SCG were prepared by enzymatic digestion following a standardized protocol for rats (*Vivas et al., 2014*) but reducing the enzyme concentration in half. Isolated neurons were plated on poly-L-lysine (Cat# P1524-25MG, Sigma)-coated glass coverslips and incubated in 5% $CO_2$ at 37°C in DMEM supplemented with 10% FBS and 0.2% penicillin/streptomycin.

## Imaging of neurite growth

Cells were imaged in culture medium at 24 and 72 hr, and 1 week after isolation for neurites analysis. Images were taken using the bright field of an LSM 880 confocal microscope (Zeiss).

## Electrophysiological recordings

Voltage responses were recorded using the perforated-patch configuration in the current-clamp mode, whereas M currents were recorded using the whole-cell configuration in voltage-clamp mode. We used an Axopatch 200B amplifier coupled with an Axon Digidata 1550B data acquisition board (Molecular Devices Electrophysiology) and HEKA EPC 9 amplifier (HEKA Elektronik) to acquire the electrical signals. Patch pipettes had a resistance of 2–4 MΩ. A liquid junction potential of 4 mV was calculated using the pCLAMP 10 software and not corrected while recording. Instead, $V_{rest}$ reported in the 'Results' section was corrected during analysis. Voltage responses were sampled at 5 kHz, whereas currents were sampled at 2 kHz. For current recordings in voltage-clamp mode, cell capacitance was canceled, and series resistances of <10 MΩ were compensated by 70%. The voltage error due to any remaining series resistance is expected to be <4 mV. The bath solution (Ringer's solution) contained 150 mM NaCl, 2.5 mM KCl, 2 mM $CaCl_2$, 1 mM $MgCl_2$, 10 mM HEPES, and 8 mM glucose, adjusted to pH 7.4 with NaOH. The internal solution used to fill the whole-cell pipettes contained 175 mM KCl, 1 mM $MgCl_2$, 5 mM HEPES, 0.1 mM $K_4BAPTA$, 3 mM $Na_2ATP$, and 0.1 mM $Na_3GTP$, adjusted to pH 7.2 with KOH. For current-clamp recordings by perforated patch, 60 µM amphotericin B (Cat# A4888) was added to the pipette solution to facilitate electrical access to the cell. The bath solution was perfused at 2 ml/min, permitting solution exchange surrounding the recording cell with a time constant of 4 s. Sodium currents were recorded 2–8 hr after isolation using a low-sodium ringer.

## Protein extraction and abundance determination

Protein from sympathetic ganglia was harvested in RIPA buffer (#89900, Thermo Scientific) with Complete, Mini, EDTA-free protease inhibitor cocktail (#11836170001, Roche) for 15 min at 4°C. Post-nuclear supernatant was isolated by centrifuging for 20 min at 13,600 × $g$ at 4°C. Protein concentration was quantified on a plate reader using the Pierce BCA protein assay kit (#23225, Thermo Scientific). Gel lanes were loaded with 30 µg of total protein. Protein samples were resolved in 4–12% Bis-Tris gels under reducing conditions. Proteins were transferred onto nitrocellulose membranes (0.2 µm; #LC2000, Life Technologies) using the Mini-Bolt system (#A25977, Thermo Scientific). Membranes were blotted using rabbit anti-KCNQ2 (ab22897, Abcam, RRID:AB_775890, 1:500). Blotted bands were detected using HRP conjugated secondary antibodies goat anti-rabbit conjugated with HRP (#1706515, Bio-Rad, 1:15,000). ImageJ was used to calculate the fluorescence density of each band. The abundance of KCNQ2 was reported as normalized to total protein and relative to the abundance in tissue from young animals.

## Reagents

XE-991 (Cat# X-101), phrixotoxin-1 (Cat# STP-700), guangtoxin-1E (Cat# STG-200), retigabine dihydrochloride (Cat# D-23129), and tetrodotoxin (T-550) were obtained from Alomone Labs. Linopirdine (Cat# L-134) was obtained from Sigma.

## Data analysis and statistics

We used IGOR Pro (IGOR Software, WaveMetrics, RRID:SCR_000325), Excel (Microsoft), and Prism (GraphPad, RRID:SCR_002798) to analyze data. ImageJ (RRID:SCR_003070) was used to process images. Data were collected from independent experiments from at least three mice and are presented

as mean ± SEM. The statistical analyses were performed using the parametric Student's *t*-test when comparing two variables and an ordinary one-way ANOVA (Dunnett's multiple comparisons test) when comparing three or more variables. GraphPad was used to calculate Pearson correlation coefficients using one-tail analysis. A non-parametric statistical test (Mann–Whitney Wilcoxon) was used to test for statistical significance between the percentage of firing and non-firing neurons. p-Values<0.05 indicate statistical significance. The number of cells used for each experiment is detailed in each figure legend.

## Acknowledgements

This work was supported by grants from the US National Institutes of Health (OV: R35GM142690; CMM: R00 AG056595), the AFAR-Glenn Foundation Junior Faculty Award (CMM), and the AFAR-Sagol Geromics Award (OV). We thank Bertil Hille for his contribution to the editing and writing of this manuscript. We thank readers Jill Jensen, Duk-Su Koh, Roya Pournejati, and Charles Asbury for their feedback on the manuscript. LdlC was awarded the Weill Neurohub Fellowship for the period 2021–2023 to support this research. CM is a Freeman Hrabowski HHMI Scholar.

## Additional information

### Funding

| Funder | Grant reference number | Author |
|---|---|---|
| National Institute of General Medical Sciences | R35GM142690 | Oscar Vivas |
| National Institute on Aging | AG056595 | Claudia M Moreno |
| American Federation for Aging Research | Sagol Network GerOmics Award | Oscar Vivas |
| American Federation for Aging Research | Glenn Foundation for Junior Faculty | Claudia M Moreno |
| Weill Neurohub | Fellowship | Lizbeth de La Cruz |

The funders had no role in study design, data collection and interpretation, or the decision to submit the work for publication.

### Author contributions

Lizbeth de La Cruz, Conceptualization, Formal analysis, Investigation, Methodology, Writing – original draft, Writing – review and editing; Derek Bui, Formal analysis, Investigation; Claudia M Moreno, Conceptualization, Funding acquisition, Writing – review and editing; Oscar Vivas, Conceptualization, Resources, Formal analysis, Supervision, Funding acquisition, Investigation, Methodology, Writing – original draft, Project administration, Writing – review and editing

### Author ORCIDs

Lizbeth de La Cruz ● https://orcid.org/0000-0003-1243-2276
Oscar Vivas ● https://orcid.org/0000-0002-0964-385X

### Ethics

Animals were handled according to the approved institutional care and use committee (IACUC) protocol (#4472-01) of the University of Washington.

Reviewer #1 (Public review): https://doi.org/10.7554/eLife.91663.4.sa1
Reviewer #2 (Public review): https://doi.org/10.7554/eLife.91663.4.sa2
Reviewer #3 (Public review): https://doi.org/10.7554/eLife.91663.4.sa3
Author response https://doi.org/10.7554/eLife.91663.4.sa4

# Additional files

## Supplementary files
• MDAR checklist

## Data availability
Data generated and analyzed is available in Dryad Digital Repository.

The following dataset was generated:

| Author(s) | Year | Dataset title | Dataset URL | Database and Identifier |
|-----------|------|---------------|-------------|-------------------------|
| de la Cruz L, Bui D, Moreno CM, Vivas O | 2024 | Sympathetic Motor Neuron Dysfunction is a Missing Link in Age-Associated Sympathetic Overactivity | https://doi.org/10.5061/dryad.z08kprrpp | Dryad Digital Repository, 10.5061/dryad.z08kprrpp |

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
