## [Editor Report · eLife assessment]

This **important** study describes changes in excitability in motor neurons of the peripheral autonomous nervous system during aging. The article provides **convincing** evidence indicating that sympathetic neurons from aged mice show higher excitability compared to neurons from young mice which was linked to decreased activity of KCNQ2/3 potassium channels. This research has implications for understanding the age-related changes that occur in the peripheral nervous system.

---

## [Referee Report · Reviewer #1 (Public review)]

Summary:

The authors study age-related changes in the excitability and firing properties of sympathetic neurons, which they ascribe to age-related changes in the expression of KCNQ (Kv7, "M-type") K+ currents in rodent sympathetic neurons, whose regulation by GPCRs has been most thoroughly studied for over 40 years.

Strengths:

The strengths include the rigor of the current-clamp and voltage-clamp experiments and the lovely, crisp presentation of the data, The separation of neurons into tonic, phasic and adapting classes is also interesting, and informative. The ability to successfully isolate and dissociate peripheral ganglia from such older animals is also quite rare and commendable! There is much useful detail here.

Weaknesses:

Whereas the description of the data are very nice, and useful, the manuscript does not provide much in the way of mechanistic insights. As such, the effect is more of an epi-phenomenon of unclear insight, and the authors cannot ascribe changes in signaling mechanisms, such as that of M1 mAChRs to the phenomena that is supported by data.

Comments on latest version:

I do not have any additional issues to be addressed by the authors.

---

## [Referee Report · Reviewer #2 (Public review)]

Summary:

This research provides compelling and detailed evidence showing that aging influences intrinsic membrane properties of peripheral sympathetic motor neurons, which become hyperexcitable. The authors found that sympathetic motor neurons from old mice exhibit increased firing rates (spontaneous and evoked), more depolarized membrane resting potential, and increased rheobase. Furthermore, the study investigates cellular mechanisms underlying age-associated hyperexcitability and shows solid evidence supporting that a decreased activity of KCNQ2/3 channels during aging is a major contributor to the increased excitability of sympathetic old neurons. The conclusions of this paper are supported by the data.

Strengths:

Detailed and rigorous analysis of electrical responses of peripheral sympathetic motor neurons using electrophysiology (perforated patch and whole-cell recordings). The study identifies a decreased KCNQ2/3 current as a cellular mechanism behind age-induced hyperexcitability in sympathetic motor neurons.

Weaknesses:

The revised version of the manuscript has addressed all my concerns.

---

## [Referee Report · Reviewer #3 (Public review)]

This revised study described changes in membrane excitability and Na+ and K+ current amplitudes of sympathetic motor neurons in culture. The findings indicate that neurons isolated from aged animals show increased membrane excitability manifested as increased firing rates in response to electrical stimulation and changes in related membrane properties including depolarized resting membrane potential, increased rheobase, and spontaneous firing. By contrast, neuron cultures from young mice show little to no spontaneous firing and relatively low firing rates in response to current injection. These changes in excitability correlate with reductions in the magnitude of KCNQ currents in neurons cultured from aged mice compared to neurons from cultured from young mice. The authors conclude that aging promotes hyperexcitability of sympathetic motor neurons through changes in KCNQ channels.

---

## [Author Response]

The following is the authors’ response to the previous reviews.

**Recommendations for the authors**:Please make corrections as suggested by reviewer 1 to improve the manuscript. Specifically, reviewer 1 suggests making changes to p values in Figure 5, and the importance of citing original scholarly works related to effects of increase in excitability of sympathetic neurons by M1 receptors, and the terminology for M currents and KCNQ currents. These changes will improve the manuscript and are strongly recommended.The section dealing with Aging Reduces KCNQ currents seems to contain a lot of extraneous information especially in the last part of the long paragraph and this section should be rewritten for improved clarity and - the implications or lack thereof - of the correlation of KCNQ with AP firing rates. The apparent lack of correlation between KCNQ current and KCNQ2 protein needs to be better explained. This is a central part of the study and this result undercuts the premise of the paper. Additionally, the poor specificity of Linordipine for KCNQ should be pointed out in the limitations.Finally, the editor notes that the author response should not contain ambiguities in what was addressed in the revision. In the original summary of consolidated revisions that were requested, one clearly and separately stated point (point 4) was that experiments in slice cultures should be strongly considered to extend the significance of the work to an intact brain preparation. The author response letter seems to imply that this was done, but this is not the case. The author response seems to have combined this point with another separate point (point 3) about using KCNQ drugs, and imply that all concerns were addressed. Authors should be clear about what revisions were in fact addressed.Summary of recommendations from the three reviewers:Please make corrections as suggested by reviewer 1 to improve the manuscript.Specifically, reviewer 1 suggests making changes to p values in Figure 5,

As a team, we have decided to keep p values. Here is our rationale:

Our lab favors reporting *p*-values for all statistical comparisons to help readers identify what we consider statistically significant. We color-coded the p-values, with red for p-value < 0.05 and black for p-value > 0.05. As a reader, seeing a p-value=0.7 allows me to know that the authors performed an analysis comparing these conditions and found the mean not to be different. Not presenting the p-value makes me wonder whether the authors even analyzed those groups. We value the ability to analyze the data by seeing all p-values than not being distracted by non-significant p-values.

and the importance of citing original scholarly works related to effects of increase in excitability of sympathetic neurons by M1 receptors, and the terminology for M currents and KCNQ currents. These changes will improve the manuscript and are strongly recommended.

We cited original papers on that area and changed the terminology for M current. I kept KCNQ when referring to the channel protein or abundance.

The section dealing with Aging Reduces KCNQ currents seems to contain a lot of extraneous information especially in the last part of the long paragraph and this section should be rewritten for improved clarity… and - the implications or lack thereof - of the correlation of KCNQ with AP firing rates.

I separated the long paragraph in two. I also removed extraneous information in that section. It now reads:

Previous work by our group and others demonstrated that cholinergic stimulation leads to a decrease in M current and increases the excitability of sympathetic motor neurons at young ages.67-71 The molecular determinants of the M current are channels formed by KCNQ2 and KCNQ3 in these neurons.70, 76, 77 Thus, Figure 6A shows a voltage response (measured in current-clamp mode) and a consecutive M current recording (measured in voltage-clamp mode) in the same neuron upon stimulation of cholinergic type 1 muscarinic receptors. It illustrates the temporal correlation between the decrease of M current with the increase in excitability and firing of APs. This strong dependence led us to hypothesize that aging decreases M current, leading to a depolarized RMP and hyperexcitability (Figure 6B). For these experiments, we measured the RMP and evoked activity using perforated patch, followed by the amplitude of M current using a whole-cell voltage clamp in the same cell. We also measured the membrane capacitance as a proxy for cell size. Interestingly, M current density was smaller by 29% in middle age (7.5 ± 0.7 pA/pF) and by 55% in old (4.8 ± 0.7 pA/pF) compared to young (10.6 ± 1.5 pA/pF) neurons (Figure 6C-D). The average capacitance was similar in young (30.8 ± 2.2 pF), middle-aged (27.4 ± 1.2 pF), and old (28.8 ± 2.3 pF) neurons (Figure 6E), suggesting that aging is not associated with changes in cell size of sympathetic motor neurons, and supporting the hypothesis that aging alters the levels of M current. Next, we tested the effect on the abundance of the channels mediating M current. Contrary to our expectation, we observed that KCNQ2 protein levels were 1.5 ± 0.1 -fold higher in old compared to young neurons (Figure 6F-G). Unfortunately, we did not find an antibody to detect consistently KCNQ3 channels. We concluded that the decrease in M current is not caused by a decrease in the abundance of KCNQ2 protein.

B. and - the implications or lack thereof - of the correlation of KCNQ with AP firing rates.

I am not sure to understand the request in the section on the correlation of KCNQ with AP firing rate. I divided the long paragraph.

The apparent lack of correlation between KCNQ current and KCNQ2 protein needs to be better explained. This is a central part of the study and this result undercuts the premise of the paper.

Indeed, total KCNQ2 protein abundance increases while M current decreases. We do not claim in our work that changes in excitability are caused by a reduction in the expression or density of KCNQ2 channels. On the contrary, our current working hypothesis is that the reduction in M current is caused by changes in traffic, degradation, posttranslational modifications, or cofactors for KCNQ2 or KCNQ3 channels. I have modified the description in the results section and discussion to clarify this concept. We also note that the discussion section contains a paragraph discussing this discrepancy.

Additionally, the poor specificity of Linordipine for KCNQ should be pointed out in the limitations.

Thank you for the suggestion. I have added the following sentences to the Limitations section. It reads: “We want to point out that linopirdine has been reported to affect other ionic currents besides M current (Neacsu and Babes, 2010; Lamas et al., 1997). Despite this limitation, the application of linopirdine to young sympathetic motor neurons led to depolarization and firing of action potentials.”

Finally, the editor notes that the author response should not contain ambiguities in what was addressed in the revision. In the original summary of consolidated revisions that were requested, one clearly and separately stated point (point 4) was that experiments in slice cultures should be strongly considered to extend the significance of the work to an intact brain preparation. The author response letter seems to imply that this was done, but this is not the case. The author response seems to have combined this point with another separate point (point 3) about using KCNQ drugs, and imply that all concerns were addressed. Authors should be clear about what revisions were in fact addressed.

We apologize for this omission. After reviewing this comment, I realized I did not respond to the Major points in the section of the Recommendations for the authors from Reviewer 3. We missed that entire section. Our previous responses addressed the Public review of Reviewer 3. When doing so, we did not separate the sentences, omitting the request to perform the experiment in slices.

The proposed experiments will require an upward microscope coupled to an electrophysiology rig; unfortunately, we do not have the equipment to do these experiments. We agree that our findings need to be tested in intact preparations to understand how the hyperactivity of sympathetic motor neurons affects systemic responses and the function of controlling organ function. This is a crucial step to move the field forward. Our laboratory is trying to find the appropriate experimental design to address this problem. We believe we must go beyond redoing these experiments in slices.

**Reviewer #1 (Recommendations For The Authors):**
(1) The significance values greater than p < 0.05 do not add anything and distract focus from the results that are meaningful. Fig. 5 is a good example. What does p = 0.7 mean? Or p = 0.6? Does this help the reader with useful information?

We thank Reviewer 1 for raising this question. We have attempted different versions of how we report *p* values, as we want to make sure to address rigor and transparency in reporting data.

Our lab favors reporting *p*-values for all statistical comparisons to help readers identify what we consider statistically significant. We color-coded the p-values, with red for p-value < 0.05 and black for p-value > 0.05. As a reader, seeing a p-value=0.7 allows me to know that the authors performed an analysis comparing these conditions and found the mean not to be different. Not presenting the p-value makes me wonder whether the authors even analyzed those groups. We value the ability to analyze the data by seeing all p-values than not being distracted by non-significant p-values.

(2) Fig. 1 is not informative and should be removed.

Although we agree with the reviewer that this figure is not informative, it was created to guide the reader in identifying the problem addressed in our manuscript in the physiological context. Our colleagues who read the first drafts of the manuscript recommended this, so we prefer to keep the figure.

(3) The emphasis on a particular muscarinic agonist favored by many ion channel physiologists, oxotremorine, is not meaningful (lines 192, 198). The important point is stimulation of muscarinic AChRs, which physiologically are stimulated by acetylcholine. The particular muscarinic agonist used is unimportant. Unless mandated by eLife, "cholinergic type 1 muscarinic receptors" are usually referred to as M1 mAChRs, or even better is "Gq-coupled M1 mAChRs." I don't think that Kruse and Whitten, 2021 were the first to demonstrate the increase in excitability of sympathetic neurons from stimulation of M1 mAChRs. Please try and cite in a more scholarly fashion.

A) We have modified lines 192 and 198, removing the mention of oxotremorine.

B) We have modified the nomenclature used to refer to cholinergic type 1 muscarinic receptors.

C) We cited references on the role of M current on sympathetic motor neuron excitability.

(4) The authors may want to use the term "M current" (after defining it) as the current produced by KCNQ2&3-containing channels in sympathetic neurons, and reserve "KCNQ" or "Kv7" currents as those made by cloned KCNQ/Kv7 channels in heterologous systems. A reason for this is to exclude currents KCNQ1-containing channels, which most definitely do not contribute to the "KCNQ" current in these cells. I am not mandating this, but rather suggesting it to conform with the literature.

Thank you for the suggestion. I have modified the text to use the term M current. I maintained the use of KCNQ only when referring to KCNQ channel, such as in the section describing the abundance of KCNQ2.

(5) The section in the text on "Aging reduces KCNQ current" is confusing. Can the authors describe their results and their interpretation more directly?(6) Please explain the meaning of the increase in KCNQ2 abundance with age in Fig. 6G. How is this increase in KCNQ2 expression consistent with an increase in excitability? The explanation of "The decrease in KCNQ current and the increase in the abundance of KCNQ2 protein suggest a potential compensatory mechanism that occurs during aging, which we are actively investigating in an independent study." is rather odd, considering that the entire thesis of this paper is that changes in excitability and firing properties are underlied by changes in KCNQ2/3 channel expression/density. Suddenly, is this not the case?? What about KCNQ3? It would be very enlightening if the authors would just quantify the ratio of KCNQ2:KCNQ3 subunits in M-type channels in young and old mice using simple TEA dose/response curves (see Shapiro et al., JNS, 2000; Selyanko et al., J. Physiol., Hadley et al., Br. J. Pharm., 2001 and a great many more). It is also surprising that the authors did not assess or probe for differences in mAChR-induced suppression of M current between SCG neurons of young and old mice. This would seem to be a fundamental experiment in this line of inquiry.

We have divided this paragraph in sections.

A. Please explain the meaning of the increase in KCNQ2 abundance with age in Fig. 6G. How is this increase in KCNQ2 expression consistent with an increase in excitability? The explanation of "The decrease in KCNQ current and the increase in the abundance of KCNQ2 protein suggest a potential compensatory mechanism that occurs during aging, which we are actively investigating in an independent study." is rather odd, considering that the entire thesis of this paper is that changes in excitability and firing properties are underlied by changes in KCNQ2/3 channel expression/density. Suddenly, is this not the case??

Our interpretation is that the decrease in M current is not caused by a decrease in the abundance of KCNQ (2) channels. We do not claim that changes in excitability are caused by a reduction in the expression or density of KCNQ2 channels. On the contrary, our working hypothesis is that the reduction in M current is caused by changes in traffic, degradation, posttranslational modifications, or cofactors for KCNQ2 or KCNQ3 channels. We have modified the description in the results section to clarify this concept. “We concluded that the decrease in M current is not caused by a decrease in the abundance of KCNQ2 protein.”

B. What about KCNQ3?

Unfortunately, we did not find an antibody to detect KCNQ3 channels. I have added a sentence to state this.

C. KCNQ2: KCNQ3 subunits in M-type channels in young and old mice using simple TEA dose/response curves.

Our laboratory is working to deeply understand the mechanism behind the changes in M current and its regulation by mAChRs in young and old ages. However, it is part of different research to attend to the complexity of the question. We think pharmacology experiments are insufficient to understand the question's complexity as we described in the next answer.

D. It is also surprising that the authors did not assess or probe for differences in mAChR-induced suppression of M current between SCG neurons of young and old mice. This would seem to be a fundamental experiment in this line of inquiry.

As mentioned, our laboratory is working to understand the mechanism behind M current and its regulation in young and old ages deeply. Our preliminary data show that M currents recorded in old neurons show two behaviors with the activation of mAChR: (1) they do not respond (blue line), or (2) they show a smaller and slower current inhibition than young neurons (red line). This data shows the complexity of the mechanism behind the M current in old neurons where changes in basal levels of PIP2, phospholipids metabolism, KCNQ2/3 changes in traffic/degradation, and M current pharmacology need to be addressed together for a proper interpretation. Showing only one part of this set of experiments in this article may lead to misinterpretation of results.

**Author response image 1. sa4fig1:** 

(7) Why do the authors use linopirdine instead of XE-991? Both are dirty drugs hardly specific to KCNQ channels at 25 uM concentrations, but linopirdine less so. The Methods section lists the source of XE991 used in the study, not linopirdine. Is there an error?A. Why do the authors use linopirdine instead of XE-991?

We use linopiridine with the experimental goal of observing the recovery phase during the washout. The main difference between the effects of XE991 and linopiridine on Kv7.2/3 is associated with the recovery phase. Currents under XE991 treatment recover 30% after 10 min compared to 93.4% with linopiridine in expression systems at -30 mV (Greene DL et al., 2017, J Pharmacol Exp Ther). After validation of KCNQ2/3 inhibition by linopirdine (IC50 value of 2.4 µM), we found linopirdine the most appropriate drug for our experiments.

Unfortunately, we were not able to observe a recovery in our experiments. The limited recovery after washout may be associated with the membrane potential of our conditions (-60 to -50 mV).

B. Both are dirty drugs hardly specific to KCNQ channels at 25 uM concentrations, but linopirdine less so.

We understand the concern of the reviewer. The specificity of XE-991 and linopiridine is not absolute. Linopiridine has been reported to activate TRPV1 channels (EC50 = 115 µM, Neacsu and Babes, 2010, J Pharmacol Sci) or nicotinic acetylcholine receptors and GABA-induced Cl- currents (EC50 = 7.6 µM and 8.1 µM respectively; Lamas et al, 1997, Eur J Neurosci).

To clarify this limitation in the article, we have added the following sentence in the section Limitations and Conclusions. “We want to point out that linopirdine has been reported to affect other ionic currents besides M current (Neacsu and Babes, 2010; Lamas et al., 1997). Despite this limitation, the application of linopirdine to young sympathetic motor neurons led to depolarization and firing of action potentials.”

C. The Methods section lists the source of XE991 used in the study, not linopirdine. Is there an error?

Thank you for pointing out this. We have added information for both retigabine and linopirdine in the Methods section; both were missing.

(8) Can the authors use a more scientific explanation of RTG action than "activating KCNQ channels?" For instance, RTG induces both a negative-shift in the voltage-dependance of activation and a voltage-independent increase in the open probability, both of which differing in detail between KCNQ2 and KCNQ3 subunits. The authors are free to use these exact words. Thus, the degree of "activation" is very dependent upon voltage at any voltages negative to the saturating voltages for channel activation.

We have modified the text to reflect your suggestion. Thank you.

(9) Methods: did the authors really use "poly-l-lysine-coated coverslips?" Almost all investigators use poly-D-lysine as a coating for mammalian tissue-culture cells and more substantial coatings such as poly-D-lysine + laminin or rat-tail collagen for peripheral neurons, to allow firm attachment to the coverslip.

That is correct. We used poly-L-lysine-coated coverslips. Sympathetic motor neurons do not adhere to poly-D-Lysine.

(10) As a suggestion, sampling M-type/KCNQ/Kv7 current at 2 kHz is not advised, as this is far faster than the gating kinetics of the channels. Were the signals filtered?

Signals were not filtered. Currents were sampled at 2KHz. Our conditions are not far from what is reported by others. Some sample at 10KHz and even 50 KHz. Others do not report the sample frequency.

**Reviewer #2:**
Weaknesses:None, the revised version of the manuscript has addressed all my concerns.We are very appreciative and glad that our responses satisfied your previous concerns.
**Reviewer #3:**
The main weakness is that this study is a descriptive tabulation of changes in the electrophysiology of neurons in culture, and the effects shown are correlative rather than establishing causality.

In the previous revision, Reviewer 3 wrote: “It is difficult to know from the data presented whether the changes in KCNQ channels are in fact directly responsible for the observed changes in membrane excitability.” And suggested the “use of blockers and activators to provide greater relevance.”

Attending this recommendation, we performed experiments in Fig. 8. Young neurons exposed to linopirdine depolarize membrane potential and promote action potential firing. In contrast, the old neurons treated with retigabine repolarize membrane potential and stop firing action potentials. This new set of experiments suggests age-related electrophysiological changes in old neurons are associated with changes in M current. The main finding of our article.

If Reviewer 3 refers to establishing causality between aging and a reduction in M current, I would like to emphasize that our laboratory is working toward a better understanding of the molecular mechanism of how M current is affected by aging; however, it will be part of a different article. One of our attempts was to reverse aging with rapamycin, but the previous recommendation was to remove those experiments.

… but the specifics of the effects and relevance to intact preparations are unclear.Additional experiments in slice cultures would provide greater significance on the potential relevance of the findings for intact preparations.

I apologize for missing this point in the previous revision. The proposed experiments will require an upward microscope coupled to an electrophysiology rig. Unfortunately, I do not

have the equipment to do these experiments.